# Generation of a conditional mutant knock-in under the control of the natural promoter using CRISPR-Cas9 and Cre-Lox systems

Vijay S. Thakur[ORCID][1], Scott M. Welford[ORCID][1,2]*

1 Department of Radiation Oncology, Miller School of Medicine, University of Miami, Miami, FL, United States of America, 2 Sylvester Comprehensive Cancer Center, Miller School of Medicine, University of Miami, Miami, FL, United States of America

* scott.welford@med.miami.edu

**Data Availability Statement:** All data are provided in the manuscript and its supporting information files.

## Abstract

Modulation of gene activity by creating mutations has contributed significantly to the understanding of protein functions. Oftentimes, however, mutational analyses use overexpression studies, in which proteins are taken out of their normal contexts and stoichiometries. In the present work, we sought to develop an approach to simultaneously use the CRISPR/Cas9 and Cre-Lox techniques to modify the endogenous *SAT1* gene to introduce mutant forms of the protein while still under the control of its natural gene promoter. We cloned the C-terminal portion of wild type (WT) *SAT1*, through the transcriptional stop elements, and flanked by LoxP sites in front of an identical version of *SAT1* containing point mutations in critical binding sites. The construct was inserted into the endogenous *SAT1* locus by Non-Homologous End Joining (NHEJ) after a CRISPR/Cas9 induced DNA double strand break. After validating that normal function of SAT1 was not altered by the insertional event, we were then able to assess the impact of point mutations by introduction of Cre recombinase. The system thus enables generation of cells in which endogenous WT *SAT1* can be conditionally modified, and allow investigation of the functional consequences of site specific mutations in the context of the normal promoter and chromatin regulation.

## Introduction

Genetic manipulation techniques, such as Cre-LoxP, flp-FRT, and CRISPR/Cas9, are extremely powerful tools to modify the genome as desired to decipher the functions of genes or genetic elements. Cre-LoxP technology has been successfully used for site and time specific gene manipulations in various species including mouse and mammalian cell cultures. The technique was developed in 1980s and is based on the ability of cyclization recombinase gene (*cre*) obtained from P1 bacteriophage to effect recombination between pairs of *loxP* (ATAAC TTCGTATAatgtatgcTATACGAAGTTAT) sites [1,2]. Such recombination can lead to the deletion of DNA present between two loxP sites if they are orientated in same direction, or flip the orientation of a DNA segment between two loxP sites if oriented in opposite directions. The technique can therefore be used to alter the genome as temporally desired by recombining segments of DNA.

**Funding:** This work was funded by NIH grant R01CA187053 to SMW. The funders had no role in study design, data collection and analysis, decision to publish, or preparation of the manuscript.

**Competing interests:** The authors have declared that no competing interests exist.

CRISPR/Cas9 is a more recent genome editing technology that was discovered from a bacterial self-defense system against viruses or plasmids. When viruses infect bacteria, the bacteria can capture a small piece of viral DNA and store them interspaced by short palindromic repeats in the form of an array known as Clustered Regularly Interspaced Short Palindromic Repeats (CRISPR). The array serves as memory for bacteria for future attacks by similar viruses. When exposed to the virus or DNA again, the bacteria use the information to synthesize non-coding crRNA, which bind to the foreign DNA with similar sequence and cause double strand breaks (DSBs) using CRISPR-associated protein 9 (Cas9) to inactivate them. As Cas9 requires a specific element named the Protospacer-associated motif (PAM) on the 3' end of the DNA sequence in order to bind to DNA and cut, the viral DNA pieces are evidently not randomly selected by the bacterial defense mechanism. This uniqueness of the CRISPR bacterial defense system has been exploited in CRISPR/Cas9 technology to induce DSBs at specific sites in the genomes of higher organisms to allow genome manipulation [3–5]. DSBs in eukaryotic organisms, for example, normally initiate DNA repair via non-homologous end joining (NHEJ) or Homology Directed Repair (HDR). NHEJ typically leads to short insertion/deletion (indels) near the cutting site, which if in a coding region of a gene or a critical genomic element, can disrupt function. On the other hand, introduction of DNA with homologous sequences near the cut site can allow insertion of a DNA segment of interest. Being so simple, versatile and precise, CRISPR/Cas9 has been quickly adopted as a preferred molecular tool in elucidating gene functions. The technique is already revolutionizing the field of biomedical research and gene therapy.

It has been reported that traditional site-specific genetic manipulation tools, like Cre/LoxP, and CRISPR/Cas9 technology can be combined to modify genes to accelerate conditional gene targeting [6,7]. In the present work, we have combined CRISPR/Cas9 and Cre/LoxP, and successfully created a CRISPR-Cas9-LoxP system to replace a part of the endogenous *SAT1* gene with an *in vitro* reconstructed wild type/mutant cassette that can be switched upon Cre expression. SAT1 (Spermidine/spermine N1-acetyltransferase 1) is the rate limiting enzyme in polyamine catabolism. We recently identified a novel role for SAT1 in gene regulation [8], which raised questions about the known, existing functions of SAT1 in its new role in transcription. Using the dual CRISPR-Cas9-LoxP system, we successful created a knock-in cassette to manipulate *SAT1* between wild type or mutant, enabling study of the importance of specific binding pockets and gain molecular insights into the normal biology of *SAT1*.

## Methods

### Construction of insert

SAT1 is a 6 exon gene spanning 3,400 nucleotides on the X chromosome (Fig 1A and S1 Fig). To construct a DNA insert to be incorporated into the *SAT1* gene to make a conditional knock-in allele, 4 different pieces of the *SAT1* gene were cloned: Part 1: approximately 2400 bp of *SAT1* starting upstream of the transcriptional start site through intron 3, with a KpnI restriction site on the 5' and an XhoI restriction site on the 3'; Part 2A: exons 4–6 with a LoxP sequence following the XhoI restriction site on the 5' end, and a ClaI restriction site on the 3' end; Part 2B: exons 4–6 including point mutations and a LoxP site following a ClaI restriction site on the 5' end, and an EcoRI restriction site on the 3' end; Part 3: from exon 6 to approximately 1000 bp downstream with EcoR1 and NotI sites on the 5' and 3' ends, respectively (S2 Fig). The parts were amplified with PCR using primers containing the desired restriction enzyme sites and LoxP sites (see Table 1 for primer sequences). The sequences of all of the amplified fragments of the gene were verified by Sanger sequencing. Each amplified piece was cloned into pBSKSII+. Two sites corresponding to amino acids 101 and 152 were mutated in

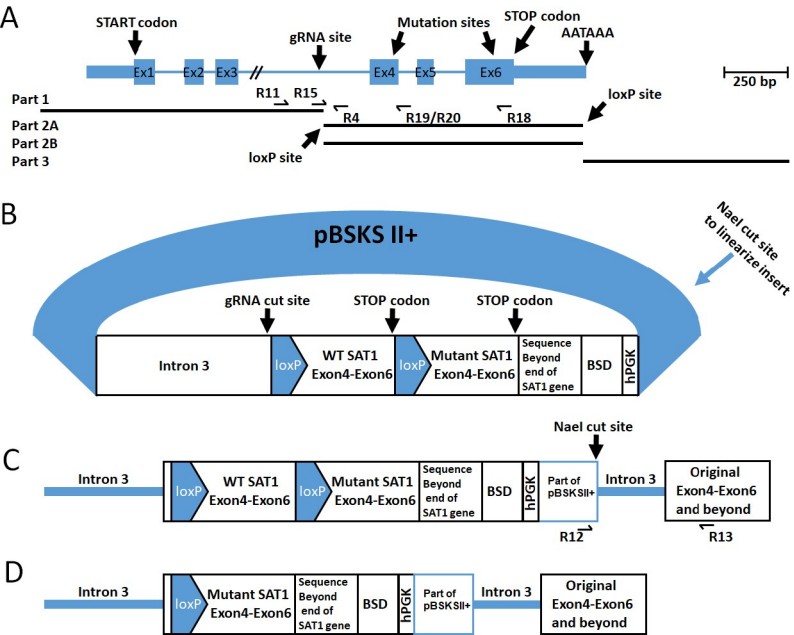

**Fig 1. Construction of insert.** A. Schematic of the *SAT1* locus, indicating exons and introns, the gRNA site location, the mutation sites, the START and STOP codons, and the transcriptional stop AATAAA sequence. Below are the four PCR 'parts' and the location of the loxP sites and the primers used for sequence validation. B. The engineered *SAT1* insert in pBSSKII+ plasmid with features indicated. C. Schematic of the *SAT1* locus after CRISPR/Cas9 mediated incorporation of the target construct. D. Schematic of the *SAT1* locus after Cre recombination. BSD = Blasticidin S deaminase; hPGK = human 3-phosphoglycerate kinase promoter.

Part 2B in pBSKSII+ using site directed mutagenesis to obtain a plasmid containing E152K and R101A point mutations (Part 2-DM), or only the E152K single mutation (Part 2-SM). The different parts were then combined to obtain the final construct. In order to make the vector blasticidin selectable, the blasticidin gene under the PGK promoter was amplified from pLX304 and inserted on the 3' side of the constructed insert enclosed within NotI restriction sites (Fig 1). Finally, the plasmid was linearized by cutting with NaeI and purified before transfection.

## Design and synthesis of sgRNA

An sgRNA was designed using the Optimized CRISPR design-site (https://chopchop.cbu.uib. no). The criteria for gRNA selection were high cutting efficiency, low off site targets, and cutting of DNA at least 150 bases upstream of exon 4. We used the Guide-it sgRNA *In Vitro* Transcription Kit (Takara Bio USA, Inc.) to produce the sgRNA. Briefly, first a forward PCR primer was designed as a DNA template for sgRNA containing our target sequence as per manufacturer's protocol. By performing a PCR reaction with the included Guide-it Scaffold Template and a primer we designed, the DNA template that contains the sgRNA-encoding sequence under the control of a T7 promoter was generated. This DNA template was *in vitro* transcribed to sgRNA and purified. The transcribed sgRNA was analyzed for its cleavage efficiency of an SAT1 DNA segment which contains the sgRNA next to a PAM sequence on its 3' end.

## Transfection

U87MG cells were plated in 6 well plates at a density of $2.5 \times 10^5$ per well. 720 ng of insert linearized by cutting with the <u>NaeI</u> restriction enzyme and purified using a gel extraction kit, 1.5μg

**Table 1. Primers used in the study.**

| Purpose | Sequence | Restriction site |
|---|---|---|
| Amplify Part 1 | F: 5'-GCAGGTACCCCCGGATCACACTTTGAGAA-3' | KpnI |
| Amplify Part 1 | R: 5'-CAGCTCGAGGACTGGCCACTCTGCAGTCT-3' | XhoI |
| Amplify Part 2A | F: 5'-GCACTCGAGATAACTTCGTATAGCATACATTATACGAAGTTATCACAGTTGTAGCCTGACTTCAGTG-3' | XhoI |
| Amplify Part 2A | R: 5'-CAGATCGATAAAGAACTAGTAAAAATGCTTACACCAAAC-3' | ClaI |
| Amplify Part 2B | F: 5'-GCAATCGATATAACTTCGTATAGCATACATTATACGAAGTTATCACAGTTGTAGCCTGACTTCAGTG-3' | ClaI |
| Amplify Part 2B | R: 5'-CAGGAATTCAAAGAACTAGTAAAAATGCTTACACCAAAC -3' | EcoRI |
| Amplify Part 3 | F: 5'-GCAGAATTCATTTCTTACACATCTTTCTTGCTGTT-3' | EcoRI |
| Amplify Part 3 | R: 5'-CAGGCGGCCGCTCTTGATTTTCAATGTTTCATAATCC-3' | NotI |
| Amplify BSD | F: 5'-GCAGCGGCCGCTTCAGACCCACCTCCCAAC-3' | NotI |
| Amplify BSD | R: 5'-CAGGCGGCCGCTGGATCTCTGCTGTCCCTGT-3' | NotI |
| Site specific mutation R101A | F: 5'-CTTCGTGATGAGTGATTATGCAGGTACGATTGAGTTCGG-3' | |
| Site specific mutation R101A | R: 5'-CCGAACTCAATCGTACCTGCATAATCACTCATCACGAAG-3' | |
| Site specific mutation E152K | F: 5'-GATCTGTCCAGTGAAAAGGGTTGGAGACTG-3' | |
| Site specific mutation E152K | R: 5'-CAGTCTCCAACCCTTTTCACTGGACAGATC-3' | |
| Amplify SAT1 sgRNA site | F: 5'-GGCGGGGAGGTAACTAAAAG-3' | |
| Amplify SAT1 sgRNA site | R: 5'-CCACTGCTGGATGATCTCAC-3' | |
| Clone test: R4 | 5'-GCCATGGCTGTCTCATGATT-3' | |
| Clone test: R11 | 5'-GTTGTCTGGGTGGTTGCTTT-3' | |
| Clone test: R12 | 5'-CTGGCGTAATAGCGAAGAGG-3' | |
| Clone test: R13 | 5'-CCTCTGCTCCGAACTCAATC-3' | |
| Clone test: R15 | 5'-CAGAGTGGCCAGTCCTCGAG-3' | |
| Clone test: R18 | 5'-AGCAGCACTCCTCACTCCTC-3' | |
| Clone test: R19 | 5'-CCGAACTCAATCGTACCTCT-3' | |
| Clone test: R20 | 5'-CCGAACTCAATCGTACCTGC-3' | |

of Cas9, and 360ng of sgRNA and 9uL lipofectamine RNAiMax (Thermo Fisher) reagent were used to transfect the cells.

## Clonal selection and characterization of clones

One day after transfection, cells were put under blasticidin selection. Individual clones were picked and expanded when sufficiently large. Genomic DNA from the clones was isolated. PCR was performed using primer pair F-R11 and R-4 to verify if the 5' end of the insert was next to the selected gRNA cut site; and primer pair F-R12 and R-R13 to verify if insert was at the correct place in the *SAT1* gene and in the correct orientation (see Table 1 for primer sequences). The amplified PCR products were gel purified and sequenced.

## Western blots

Westerns were performed using standard procedures by running RIPA cell lysates on denaturing polyacrylamide gels, transferring to PVDF membranes, and staining with the following antibodies: SAT1: ab105220 (Abcam); MELK: HPA017214 (Sigma); FOXM1: PA5-27144 (Thermo Fisher); β-actin: A1978 (Sigma). Quantification of the blots was performed with ImageJ, and statistical calculations by student's t tests were performed with GraphPad Prism 7.05.

### DNA gels electrophoresis

Post PCR samples were separated on 1.2% agarose gels and stained with ethidium bromide, and then visualized with a UV illuminator according to standard methods. A 1 kilobase DNA ladder was used as a size standard.

## Results

In order to test the effect of specific SAT1 mutants in the biology of polyamine metabolism, common approaches of shRNA or even CRISPR/Cas9 knockout followed by re-expression of mutant constructs are inadequate due to the challenge of replicating endogenous levels of expression. We therefore attempted to develop a CRISPR/Cas9-based knockin approach to insert appropriate mutants into the normal SAT1 locus. The U87MG glioblastoma cell line was chosen as the target cell line because it is easy to culture and is amenable to viral infections. Therefore, the SAT1 locus, as indicated in Fig 1A was used to design a knock-in scheme whereby exons 4–6 would be duplicated with the wild type exons flanked by loxP sites, and immediately followed by mutant exons 4–6. The concept is to maintain all of the regulatory elements of the endogenous locus, including the intron/exon boundaries, the transcriptional termination sequence, and any potential transcriptional regulatory elements within or down-stream of the gene by extending the construct well beyond the normal end of transcription before introducing a selectable antibiotic resistance marker (blasticidin s deaminiase) (Fig 1B).

### Cutting efficiency of the sgRNA

The sequence of intron 3 of SAT1 (tatttactattctgaactgccgtgtaaacctgacgtattcccaagtcaacataccagta-taccaataggatgtgaataatgtgtgtgttgagtttaaaaccatagcagttttgctctggcaagtaatgaaagcgttctcgcttcctgagtgt-gagctccagcagactgcagagtggccagtc) was analyzed by the CHOPCHOP gRNA design tool (https://chopchop.cbu.uib.no), and the high scoring gRNA sequence GGGAATACGTCAGGTTTACA was selected. A 56 nucleotide forward primer CCTCTAATACGACTCACTATAGGGAATAC GTCAGGTTTACAGTTTAAGAGCTATGC was designed to create the DNA template for the sgRNA. After gel electrophoresis, the sgRNA had a band that appeared the proper size of 150bp (Fig 2A). The sgRNA was tested for its cleavage efficiency at the selected sequence on inton 3 of the *SAT1* gene. The sgRNA was incubated with a fragment amplified from genomic DNA obtained from U87MG cells containing the gRNA sequence (Fig 2B) and Cas9 endonu-clease *in vitro*. The DNA fragment was cleaved into two pieces with near 100% efficiency (Fig 2C, lane 2).

### Verification of incorporation of the insert in the correct place and orientation in the *SAT1* gene

According to the design, after transfection, Cas9 will cleave endogenous *SAT1* as well as the linearized insert at the sgRNA site. Notably, U87MG cells have only one X chromosome, facili-tating the approach. The insert may be randomly or specifically inserted during NHEJ DNA repair. Insertion of DNA fragments as big as 48 kilobases using NHEJ repair has been success-fully used in the past and has shown that the percentage of cells which incorporated large frag-ments into the genome was higher than using an HDR mechanism [9], particularly as a significantly lower number of cells are in S/G2 phase of the cycle at any given time compared to the entire population. In order to confirm that the insert was placed into the *SAT1* gene at the right place and orientation, and a functional wild type *SAT1* gene remained intact, the cells were selected using blasticidin and genomic DNA was PCR amplified. Primer R11, which binds the *SAT1* gene upstream of the gRNA cut site, and R4, which binds to downstream of

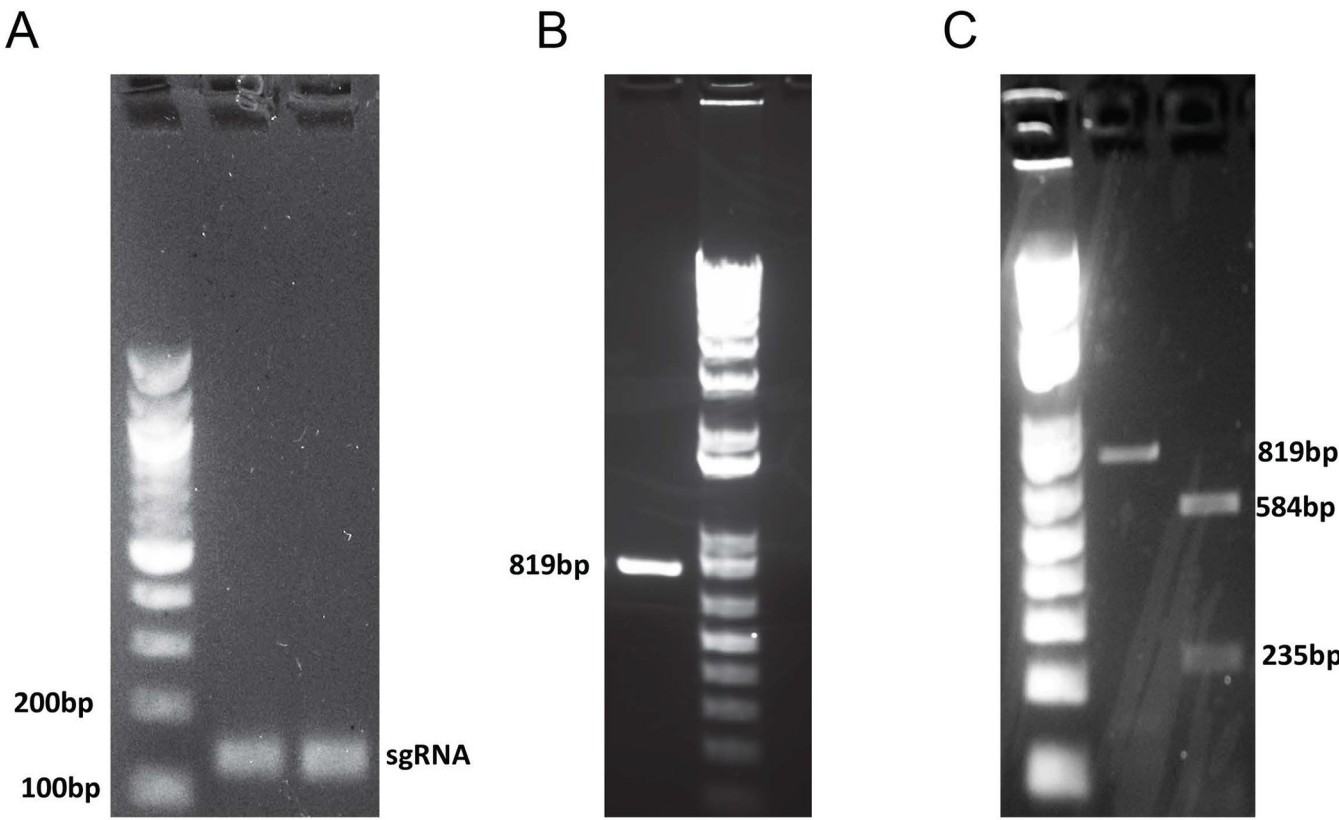

**Fig 2. *In vitro* transcription and screening of sgRNA.** A. Gel electrophoresis of the PCR product of *In Vitro* Transcription of sgRNA B. PCR product of 819-bp target fragment amplified from the *SAT1* gene containing gRNA sequence. C. The PCR amplified *SAT1* fragment, the sgRNA, and recombinant Cas9 enzyme were combined in an *in vitro* cleavage reaction according to the protocol. The reaction mixture was run on 2% Agarose gel. Lane. 1 shows the untreated target *SAT1* fragment, and Lane 2 shows the Cas9 treated, cleaved target fragments.

the gRNA cut site after the LoxP sequence on the wild type exon 4-exon6 part, were used. We also performed PCR on genomic DNA from the clones to further confirm that the insert has incorporated at the correct place using the R12 forward primer which binds to pBSKSII+ plasmid sequences remaining on the insert after linearization, and the R13 reverse primer that binds to intron 4 of *SAT1* (Fig 3). After PCR, gel purified PCR products were subjected to

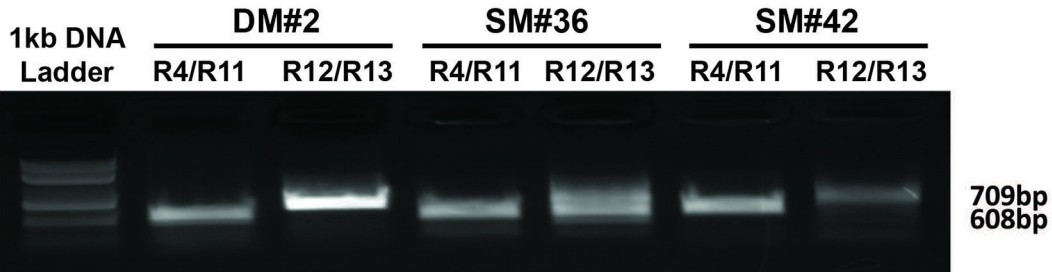

**Fig 3. Genomic PCR analysis of positive clones for verification of incorporation of the insert in the right place and orientation in the *SAT1* gene.** Primer pair R4/R11 and R12/R13 amplify the 5'-junction (608bp) and the 3'-junction (709bp) of the engineered insert integration, respectively. All amplified DNA fragments exhibited expected sizes (including some additions and deletion between the sequences of primers), indicating correct integration of the insert into endogenous *SAT1* gene via NHEJ.

DNA sequencing for confirmation of amplification of the proper fragments (S3 Fig). In order to identify the successful clones, we screened a total of 84 clones (27 for the double mutant, and 57 for the single mutant. PCR success rate was 32% (27/84), but upon further analyses to ensure all parts of the vector were incorporated, 3 of the 27 clones were deemed correct. Thus the overall success rate was 3.6%.

## Confirmation of conversion of wild type *SAT1* to mutant *SAT1* after Cre expression

In order to verify conditional conversion of wild type *SAT1* to mutant *SAT1*, cells of clones DM#2 and SM#36 were infected with either 200 multiplicity of infection (MOI) or 800 MOI control adenovirus or Cre recombinase expressing viruses for 3 days. Genomic DNA was isolated from the cells and amplified using forward primer R15 which binds to both wild type as well as mutated *SAT1* (R101A). The reverse primers used were R19 which amplify only wild type *SAT1* and not mutated *SAT1* (R101A), and R20 which will amplify mutated *SAT1* (R101A) but not wild type *SAT1*. The primers R19 and R20 have two unique nucleotides on the 3' end. Results showed that the wild type gene amplified using primer set R15/R19 in control Adeno virus treated clones but not when treated with Adeno-Cre viruses; whereas with primer set R15/R20, the wild type gene was not amplified in control adenovirus treated cells but was amplified in Adeno-Cre treated clonal cells (Fig 4A). The amplified DNA was gel extracted and sequenced to confirm the conditional conversion of wild type *SAT1* to the mutant forms. As the primers have >90% similar sequence, some amplification with R15/R19 primers in Adeno-Cre virus treated samples and with R15/R20 primers in Adeno-control samples occurred. DNA sequencing of these bands however still confirmed wild type or mutant *SAT1* in adeno-control or adeno-Cre treated samples, respectively (Fig 4B).

As mutation E152K is caused by to only one nucleotide change, unlike R101A where two nucleotides change, we were not able to design a primer which can cause differential amplification of wild type vs. the E152K mutant. Therefore, after PCR amplification using primer sets R15/R18 to amplify the region of the second mutation, DNA sequencing was performed to confirm the conversion of wild type to mutant *SAT1* (Figs 4C, 4D, 5A and 5B). DNA sequencing confirmed that expression of Cre using adenovirus was able to delete the wild type part of *SAT1* between exons 4–6, and allow replacement with mutant exons 4–6. In SM#36, Cre was also expressed from Lentivirus to achieve the same result. DNA sequencing of the DNA fragments obtained using R15/R18 primers again confirmed the conversion of wild type *SAT1* gene to the mutant form (Fig 5C and 5D).

## Effect of SAT1 mutation on its downstream targets genes

We have previously reported that *MELK* and *FOXM1* are downstream targets of SAT1 transcriptional activity, and SAT1 enzymatic activity is required for this role [8]. We wanted to examine the effect of conversion of wild type SAT1 to mutant SAT1, which causes loss of its polyamine acetylation enzymatic activity, on transcriptional activity. As an essential control experiment, we first measured the levels of each of SAT1, FOXM1, and MELK in the mutant clones compared to the parental U87MG cell line from which they were derived. In the absence of Cre recombination, we found that the levels of each of the three proteins were not significantly different (Fig 6A and 6B), suggesting we had not inadvertently disrupted the basal function of SAT1, and that targeting was sufficiently specific. We then proceeded to assess the effect of Cre recombination on FOXM1 and MELK. As see in Fig 6C and 6D, the conversion of wild type SAT1 to mutant SAT1 did not significantly alter SAT1 protein levels, but the levels of MELK and FOXM1 decreased significantly. The results confirm that the CRISPR/

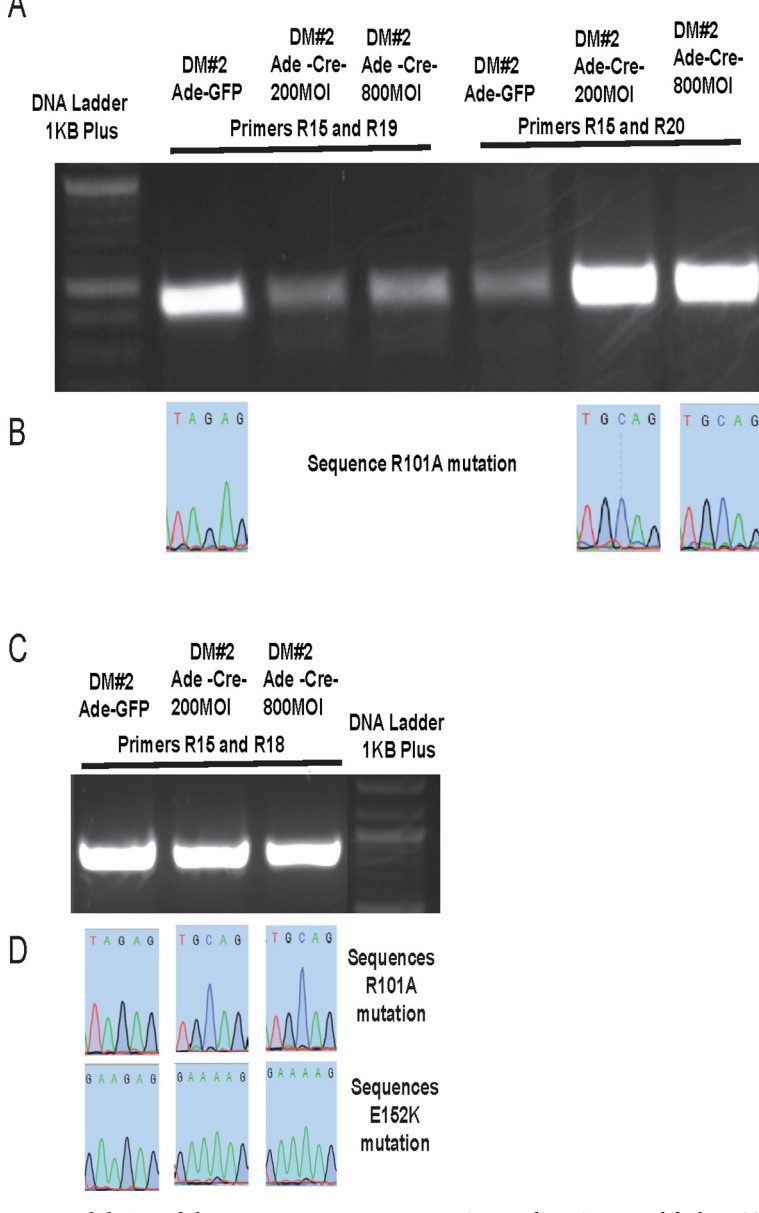

**Fig 4. Validation of clone DM#2.** A. Primer pair R15/R19 and R15/R20 amplify the WT or mutant *SAT1*, respectively. B. Sequence results of the PCR products confirm the conversion of WT codon (AGA) to mutant codon (GCA) of amino acid 101 of SAT1 protein. C. Primer pairs R15/R18 and R17/R18 amplify the WT and mutant *SAT1* at nucleotides corresponding to amino acids 101 and 152 of SAT1 protein. D. Sequence results of the PCR products confirm the conversion of WT to mutant *SAT1* gene.

Cas9-LoxP approach for generation of conditional mutant gene under the control of its own promoter is attainable and useful for studying *in vivo* gene functions.

## Discussion

In this study, by combining CRISPR-Cas9 and Cre/LoxP technologies, we successfully generated cell lines in which wild type *SAT1* can be conditionally modified to a mutant form under the control of its natural promoter. We exploited NHEJ repair to introduce an engineered

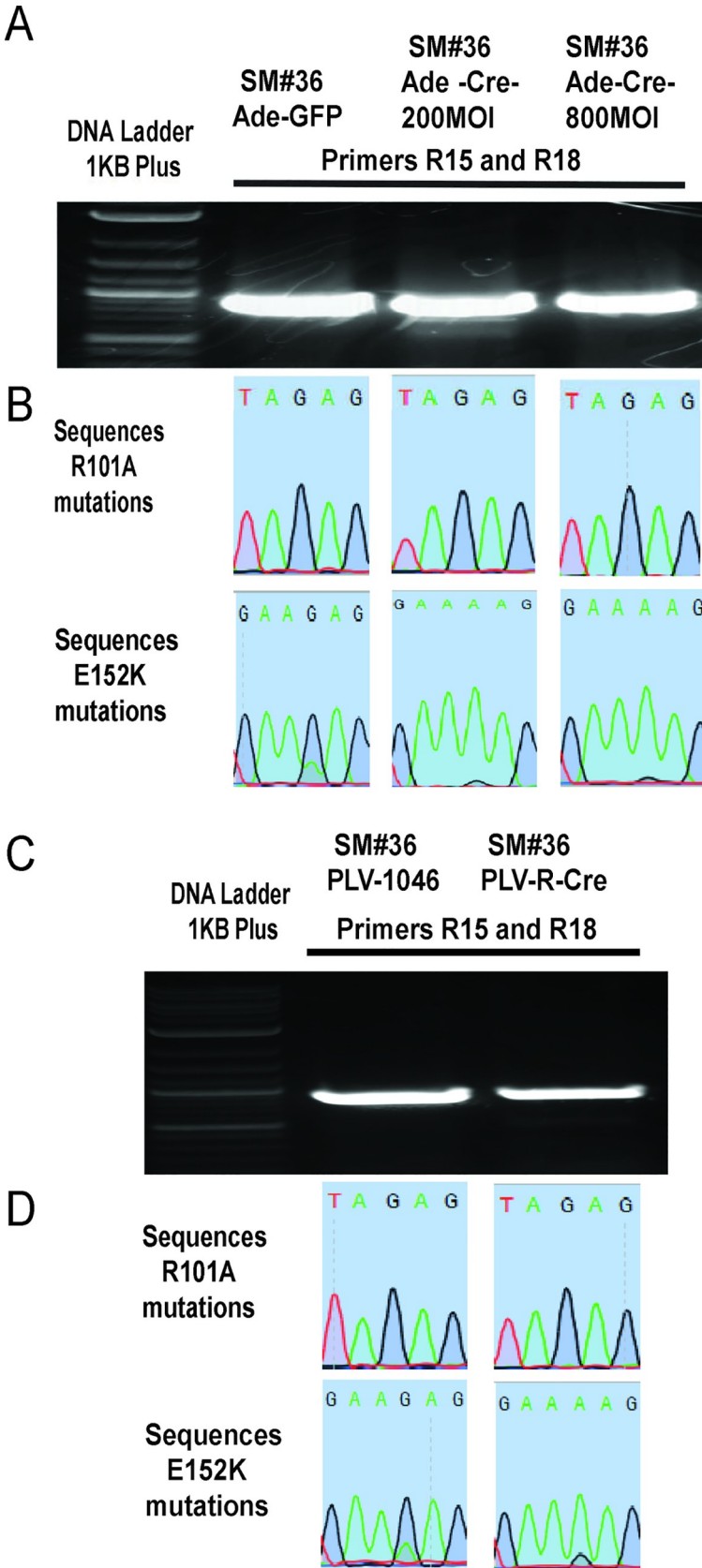

**Fig 5. Validation of clone SM#36.** A. DNA gel showing DNA amplification of WT or mutant *SAT1* using R15/R18 and R17/R18 primers. B. Sequence results of the PCR products confirm the conversion of WT codon (AAG) to mutant codon (AAA) of amino acid 152 of SAT1 protein after Cre recombination. C. DNA gel showing DNA amplification of WT or mutant *SAT1* using R15/R18 and R17/R18 primers after Cre recombination. D. Sequence results of the PCR products confirm the conversion of the WT codon (AGA) to the mutant codon (GCA) for amino acid 101; and the WT codon (AAG) to the mutant codon (AAA) for amino acid 152 of SAT1.

DNA fragment into the *SAT1* gene after induction of a CRISPR-Cas9-induced DNA double stranded break at a specific site on the *SAT1* gene. The design presented here provides a framework that could be exploited for any other gene in the genome, depending on the complexity of a given locus. For SAT1, the tools allow the dissection of function between what is known in the literature from a variety of studies and a novel function that we recently uncovered, namely as a transcriptional regulator. Given that exogenous expression of SAT1 causes general failure of protein translation and subsequent cell death [10], a knock-in strategy to avoid toxicity of overexpression was essential. Likewise, overexpression of many genes leads to artificial phenotypes. Thus, using the CRISPR-Cas9-LoxP approach allows relevant observation of endogenous biological functions.

Knocking in exogenous DNA into a specific site in the genome to maintain genomic integrity has significant experimental advantages. In the past, an HDR approach has been

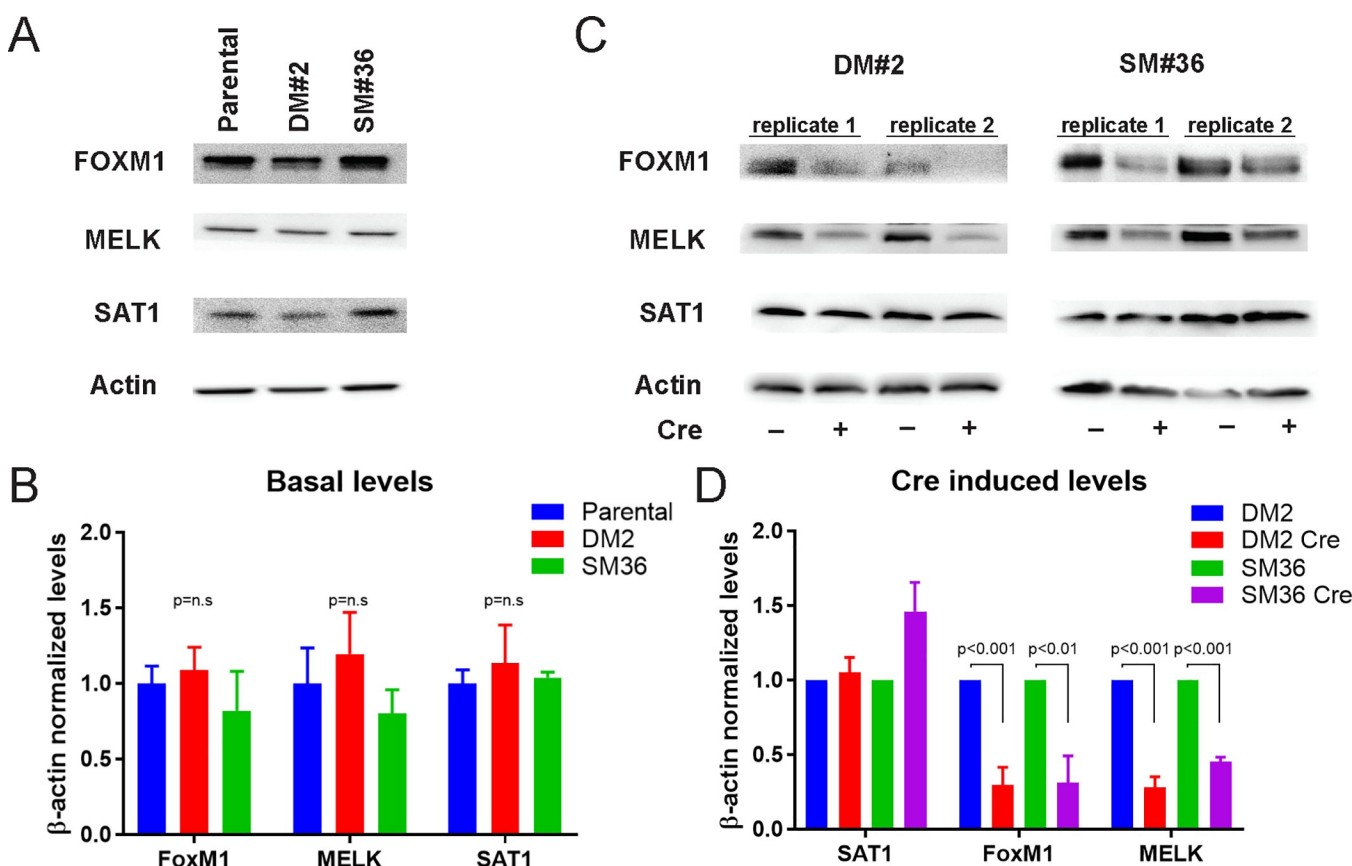

**Fig 6. Effect of SAT1 mutation on its downstream targets genes.** A. Western blot of parental U87MG and knockin clones DM#2 and SM#36 demonstrating insignificant changes to basal levels of FOXM1, MELK, and SAT1. B. Quantification of three replicate experiments. C. Western blot of Cre infected DM#2 and SM#36 showing changes in FOXM1 and MELK. D. Quantification of three replicate experiments. Uncropped and replicate westerns are shown in supplementary data.

successfully applied to introduce a fragment of DNA into a gene by exogenously providing a complimentary sequence. Homologous recombination for precise DNA insertion or replacement at a selected genomic locus has been widely used with great success in generating genetically modified cells in culture and in experimental animals [11–15]. As HDR occurs only during S/G2 phase, however, the process can result in a low efficiency. Therefore, a modified approach would be advantageous to increase the practicality of knock-in experiments. DNA repair using NHEJ takes place throughout the cell cycle and is a dominant DNA repair mechanism in mammalian cells [9]. Even in HDR based gene targeting studies, the frequency of NHEJ indels is higher than HDR mediated DNA insertion even when single strand oligonucleotides were provided for HDR-mediated gene correction [16,17]. Lin, et al. reported the capture of DNA sequences at double-strand breaks in mammalian chromosomes [18]. Orlando et al. reported successful insertion of small oligonucleotides efficiently can be inserted via NHEJ at the DSB caused by Zinc finger nucleases on genomic DNA [19]. Three years later, two studies showed that successful integration of large plasmid DNAs can be successfully integrated in the genome by NHEJ repair mechanism after simultaneous cleavage of both plasmid and genome DNAs by ZFN/TALEN nucleases [20,21]. In the genome of zebrafish and *Xenopus*, where HDR-mediated gene insertion is extremely inefficient, an NHEJ repair mechanism was successfully and efficiently used to insert DNAs at CRISPR/Cas9 generated DSBs [22–24]. A recent study showed that homology independent high knock-in events mediated by NHEJ, were able to knock-in DNA as big as 34 kilobases with much higher efficiency than using HDR [9]. In the present study we used an NHEJ approach, because the insert had no region of homology, to knock-in our DNA insert into *SAT1* gene.

No gene editing technology (to date) is perfect, and some limitation must be considered. In the present study, we took advantage of the fact that SAT1 is localized to the X chromosome, and using cells derived from a male glioblastoma patient meant that only a single locus needed to be edited. We cannot at present assume that the success rate would have been the same if two loci were targeted for an autosomal gene. That said, CRISPR/Cas9 approaches frequently knockout two loci without tremendous reductions in efficiency [6], and it is likely that a similar strategy would work for genes not on the X chromosome in a male cell line. An additional limitation, which is common for all template based knock-in approaches, could be the length of insert. We were fortunate in the case of SAT1 to be targeting to amino acids relatively close to each other. Should one desire to alter residues at greater distances, there is no reason to think a second round of CRISPR/Cas9 would be any more challenging than the first, and one could imagine significantly more complicated schemes that would still be controlled by Cre in the end.

Results of this study confirm that CRISPR/Cas9-coupled NHEJ repair can provide a valuable path for efficient knock-in of large DNA segments in human cells. Wild type SAT1 can be conditionally converted to a mutant version while still under the control of its natural promoter by supplying Cre recombinase at the desired time. The system can thus be used to study and identify the function of enzymatic activity of SAT1 gene in regulating its downstream pathways and providing better understanding of its mechanism in genomic function. The approach can also be expanded to study other genes in cells.

## Supporting information

**S1 Fig. Sequence of SAT1 gene.** Introns are depicted small letters and exons in capital letters.
(PPTX)

**S2 Fig. Parts of the SAT1 CRISPR/Cas9 gene editing construct.**
(PPTX)

**S3 Fig. Sequences of ligation sites in clones.** Addition or deletion on 5' and 3' end of the insert on DNA sequence.
(PPTX)

**S4 Fig. Raw gel images.**
(PPTX)

**S5 Fig. Raw and triplicate western images in different exposures.**
(PPTX)

## Acknowledgments

We wish to thank all members of the Welford Lab for helpful discussions to the work.

## Author Contributions

**Conceptualization:** Vijay S. Thakur, Scott M. Welford.

**Formal analysis:** Scott M. Welford.

**Funding acquisition:** Scott M. Welford.

**Investigation:** Vijay S. Thakur.

**Methodology:** Vijay S. Thakur.

**Project administration:** Scott M. Welford.

**Supervision:** Scott M. Welford.

**Writing – original draft:** Vijay S. Thakur.

**Writing – review & editing:** Scott M. Welford.

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
