## [Decision Letter · Decision Letter 0]

17 Jun 2020

PONE-D-20-13689

Generation of a conditional mutant knock-in under the control of the natural promoter using CRISPR-Cas9 and Cre-Lox systems.

PLOS ONE

Dear Dr. Welford,

Thank you for submitting your manuscript to PLOS ONE. After careful consideration, we feel that it has merit but does not fully meet PLOS ONE’s publication criteria as it currently stands. Therefore, we invite you to submit a revised version of the manuscript that addresses the points raised during the review process.

There are both experimental, statistical and textual issues that must be completed.Experimental and Statistical.Present the raw data for Western blots in supplementary experiments.Provide a link to all of the sequencing data. Provide the raw data for Western blots and all other electrophoretic experiments.As noted by Reviewer 2, the Western blots of Sat1 and its targets must be compared with the original cell line.Please indicate the number of times the Western analysis was performed, quantify the data, and determine statistical significance. If the experimental has not been repeated, please conduct at least three times (including the above controls).TextualIn the Results section, 1 st paragraph what is the meaning of "score"The Results section should open with a brief overview of the method that should be shown diagrammatically in a new Figure 1.The Materials and Methods are incomplete and lack methods including electrophoretic and blotting techniques such as Western analysis as sequencing and statistical analysis (for Westerns).The text contains multiple grammatical errors throughout.  Please correct these errors. (Examples: Abstract:"an identical region SAT1 containing";  "CRISPR/Cas9 induced DNA double break".)Please provide more detail in the current Figure 1 as suggested by Reviewer 3. Address all other issues not mentioned above from the Reviewers.There are no conflicts among the Reviewers. They present different valuable insights.The AE evaluation focused on structural aspects of the manuscript and experimental issues (as noted above).

We look forward to receiving your revised manuscript.

Kind regards,

Arthur J. Lustig, PhD

Academic Editor

PLOS ONE

Journal Requirements:

3. Please include your tables as part of your main manuscript and remove the individual files. Please note that supplementary tables (should remain/ be uploaded) as separate "supporting information" files

Reviewers' comments:

Reviewer's Responses to Questions

**Comments to the Author**

1. Is the manuscript technically sound, and do the data support the conclusions?

Reviewer #1: Yes

Reviewer #2: No

Reviewer #3: Yes

2. Has the statistical analysis been performed appropriately and rigorously? 

Reviewer #1: N/A

Reviewer #2: No

Reviewer #3: N/A

3. Have the authors made all data underlying the findings in their manuscript fully available?

Reviewer #1: Yes

Reviewer #2: Yes

Reviewer #3: Yes

4. Is the manuscript presented in an intelligible fashion and written in standard English?

Reviewer #1: Yes

Reviewer #2: Yes

Reviewer #3: Yes

5. Review Comments to the Author

Reviewer #1: 

Comments:

Authors describe an approach to introduce mutant forms of the endogenous SAT1 gene in mammalian cells by simultaneously using the CRISPR/Cas9 and Cre-Lox techniques. This approach could also be expanded to dissect functions of other genes that are important for cell growth and therefore cannot be simply deleted.

Minor issues to be addressed:

1. Are there any published approaches that combining CRISPR/Cas9 and Cre-Lox systems, either using in mammalian or in other eukaryotic cells? It would be valuable for the readers to know the differences between the existing methods and the one established in this study.

2. Please give the detail data about the efficiency and the positive rate of this method in the text.

3. Please give a clear information of “U87MG cells” in the text.

4. In figure 1, please explain “BSD” and “hPGK” in the figure legend.

Reviewer #2: The manuscript by Thakur and Welford describe a smart system to create inducible mutant alleles combining CRISPR/Cas9 and Cre/lox in one transgenesis cassette integrated in the locus of choice via NHEJ.

Several aspects of the manuscript remain a bit obscure and needs to be clarified for the efficiency of the method to be evaluated by potential other users.

1)How many clones where tested to find the two ones that have the correct integrations and are further characterized?

2)it would be much easier to evaluate the PCR results if the primer position would be indicated in a schematic like the one in fig 1B-C.

3)The western blot analysis of Sat1 and its targets should be integrated with a comparison of the same levels in the cell lines from which the transgenic ones were derived (U87MG) to show that the cassette integration doesn't affect the expression and is regulated at endogenous levels.

4)The western blots should be performed in triplicates at least and the results quantified as it's common standard nowadays. Also Uncropped blots should be made available.

Minor point

The file "paper tables" contains also the supplementary figures that are therefore duplicated.

Reviewer #3: In the current manuscript, Thakur VS et al. developed an interesting approach to simultaneously use the CRISPR/Cas9 and Cre-Lox techniques to modify the endogenous SAT1 gene to introduce mutant forms of the protein while still under the control of its natural gene promoter. The authors first inserted a construct containing both wild type and mutant Exon4-Exon6 fragments into intron 3 of the endogenous SAT1 locus by NHEJ after a CRISPR/Cas9-induced DSB. The wild-type Exon4-6 fragment was then deleted by Cre recombinase. The authors show that the SAT1 mutant cell lines had reduced FoxM1 and MELK expression. Although the manuscript is well written, several points require further clarification.

1. Supplementary Figure 1 and 2 should be summarized as a figure and put in Figure 1. The scheme should include start codon, stop codon, mutation sites etc.

2. Stop codon should be marked in both WT and mutant SAT1 in Figure 1.

3. Figure 4 and 5, the location of primers should be labeled in a scheme of the transgene.

4. Figure 4C and D are not described in the text.

5. Figure 6, additional SAT1 activity assays would strengthen the conclusion.

6. Off-target integrations occurred during the NHEJ-mediated knock-in were discussed, but not studied.

7. Limitations of this new approach should be discussed.

6. PLOS authors have the option to publish the peer review history of their article (what does this mean?). If published, this will include your full peer review and any attached files.

Reviewer #1: No

Reviewer #2: No

Reviewer #3: No

---

## [Author Response · Author response to Decision Letter 0]

21 Aug 2020

AE:

o Experimental and Statistical.

Present the raw data for Western blots in supplementary experiments.

Raw data now presented in supplementary figure 4 and 5.

Provide a link to all of the sequencing data. 

All sequencing data are presented in supplementary figure 3.

Provide the raw data for Western blots and all other electrophoretic experiments.

Raw data now presented in supplementary figure 4 and 5.

As noted by Reviewer 2, the Western blots of Sat1 and its targets must be compared with the original cell line.

New data has been added to figure 6 to address this.

Please indicate the number of times the Western analysis was performed, quantify the data, and determine statistical significance. If the experimental has not been repeated, please conduct at least three times (including the above controls).

All westerns were performed three times, quantified, and presented now in figure 6 and supplementary 5.

o Textual

In the Results section, 1 st paragraph what is the meaning of "score"

The gRNA design tool CHOPCHOP was used, and a high scoring gRNA with minimal predicted off target sites was chosen. We have clarified this in the text.

The Results section should open with a brief overview of the method that should be shown diagrammatically in a new Figure 1.

A new Figure 1A has been made, and appropriately introduced.

The Materials and Methods are incomplete and lack methods including electrophoretic and blotting techniques such as Western analysis as sequencing and statistical analysis (for Westerns).

More details were added.

The text contains multiple grammatical errors throughout. Please correct these errors. (Examples: Abstract:"an identical region SAT1 containing"; "CRISPR/Cas9 induced DNA double break".)

Completed.

Please provide more detail in the current Figure 1 as suggested by Reviewer 3. 

Completed.

Reviewer #1: 

Minor issues to be addressed:

1. Are there any published approaches that combining CRISPR/Cas9 and Cre-Lox systems, either using in mammalian or in other eukaryotic cells? It would be valuable for the readers to know the differences between the existing methods and the one established in this study.

Indeed, there is a published approach that combined CRISPR/Cas9 and Cre-Lox systems and has been cited in the paper (reference 7). However, in that study, WT gene was not modified, but an exogenous gene under an artificial promoter was introduced which can then be manipulated using Cre-Lox system. In our paper, we modified the WT SAT1 gene such that it can be converted to its mutated version by cre activation while still under its natural promoter. 

2. Please give the detail data about the efficiency and the positive rate of this method in the text.

This data has been added to the results.

3. Please give a clear information of “U87MG cells” in the text.

U87MG has been properly introduced.

4. In figure 1, please explain “BSD” and “hPGK” in the figure legend.

The abbreviations have been added to the legend.

Reviewer #2: 

1)How many clones where tested to find the two ones that have the correct integrations and are further characterized?

This data is added to the results.

2)it would be much easier to evaluate the PCR results if the primer position would be indicated in a schematic like the one in fig 1B-C.

Primer locations were added to Figure 1A

3)The western blot analysis of Sat1 and its targets should be integrated with a comparison of the same levels in the cell lines from which the transgenic ones were derived (U87MG) to show that the cassette integration doesn't affect the expression and is regulated at endogenous levels.

This data was now added to figure 6.

4)The western blots should be performed in triplicates at least and the results quantified as it's common standard nowadays. Also Uncropped blots should be made available.

All westerns were performed three times, quantified and the data are presented in figure 6 and in supplementary figure 5

Minor point

The file "paper tables" contains also the supplementary figures that are therefore duplicated.

The duplication was removed.

Reviewer #3: 

1. Supplementary Figure 1 and 2 should be summarized as a figure and put in Figure 1. The scheme should include start codon, stop codon, mutation sites etc.

A new panel was added to figure 1.

2. Stop codon should be marked in both WT and mutant SAT1 in Figure 1.

Stop codons are identified.

3. Figure 4 and 5, the location of primers should be labeled in a scheme of the transgene.

Primer locations were added to Figure 1A

4. Figure 4C and D are not described in the text.

The text was amended to include the references to Figure 4C and D

5. Figure 6, additional SAT1 activity assays would strengthen the conclusion.

We agree that activity assays would strengthen the biology behind the story here, but felt this was a bit beyond the scope of the study. As we are not claiming a biological advance in this work, the sequencing confirms the presence of the alteration. We do appreciate the comment and are applying this to future studies currently.

6. Off-target integrations occurred during the NHEJ-mediated knock-in were discussed, but not studied.

There is of course always a chance that off target editing will occur. Because we saw no changes to the endogenous levels of SAT1 before Cre recombinase activity excised the loxP cassette, we are confident that off target effects are likely minimal in our case. A comment was added to the results to address this point.

7. Limitations of this new approach should be discussed.

The discussion was amended to address limitations.

---

## [Decision Letter · Decision Letter 1]

23 Sep 2020

Generation of a conditional mutant knock-in under the control of the natural promoter using CRISPR-Cas9 and Cre-Lox systems.

PONE-D-20-13689R1

Dear Dr. Welford,

We’re pleased to inform you that your manuscript has been judged scientifically suitable for publication and will be formally accepted for publication once it meets all outstanding technical requirements.

Kind regards,

Arthur J. Lustig, PhD

Academic Editor

PLOS ONE

Additional Editor Comments (optional):

Reviewers' comments:

Reviewer's Responses to Questions

**Comments to the Author**

1. If the authors have adequately addressed your comments raised in a previous round of review and you feel that this manuscript is now acceptable for publication, you may indicate that here to bypass the “Comments to the Author” section, enter your conflict of interest statement in the “Confidential to Editor” section, and submit your "Accept" recommendation.

Reviewer #2: All comments have been addressed

Reviewer #3: All comments have been addressed

2. Is the manuscript technically sound, and do the data support the conclusions?

Reviewer #2: Yes

Reviewer #3: Yes

3. Has the statistical analysis been performed appropriately and rigorously? 

Reviewer #2: Yes

Reviewer #3: Yes

4. Have the authors made all data underlying the findings in their manuscript fully available?

Reviewer #2: Yes

Reviewer #3: Yes

5. Is the manuscript presented in an intelligible fashion and written in standard English?

Reviewer #2: Yes

Reviewer #3: Yes

6. Review Comments to the Author

Reviewer #2: The authors have addressed all my major concerns. I believe now the manuscript is sound and ready for publication. I specifically acknowledge the effort made by the authors to address some of the points raised with new experiments under the current difficult circumstances.

Reviewer #3: (No Response)

7. PLOS authors have the option to publish the peer review history of their article (what does this mean?). If published, this will include your full peer review and any attached files.

Reviewer #2: No

Reviewer #3: No

---

## [Editor Report · Acceptance letter]

24 Sep 2020

PONE-D-20-13689R1 

Generation of a conditional mutant knock-in under the control of the natural promoter using CRISPR-Cas9 and Cre-Lox systems. 

Dear Dr. Welford:

I'm pleased to inform you that your manuscript has been deemed suitable for publication in PLOS ONE. Congratulations! Your manuscript is now with our production department. 

Kind regards, 

on behalf of

Dr. Arthur J. Lustig 

Academic Editor

PLOS ONE